# Comparative Genomics Applied to Systematically Assess Pathogenicity Potential in Shiga Toxin-Producing *Escherichia coli* O145:H28

**DOI:** 10.3390/microorganisms10050866

**Published:** 2022-04-21

**Authors:** Michelle Qiu Carter, Nicole Laniohan, Chien-Chi Lo, Patrick S. G. Chain

**Affiliations:** 1Produce Safety and Microbiology Research Unit, U.S. Department of Agriculture, Agricultural Research Service, Western Regional Research Center, Albany, CA 94710, USA; nicole.laniohan@usda.gov; 2Biosecurity and Public Health Group, Los Alamos National Laboratory, U.S. Department of Energy, Bioscience Division, Los Alamos, NM 87545, USA; chienchi@lanl.gov (C.-C.L.); pchain@lanl.gov (P.S.G.C.)

**Keywords:** Shiga toxin-producing *Escherichia coli* (STEC), pangenome, pathogenicity islands, virulence genes

## Abstract

Shiga toxin-producing *Escherichia coli* (STEC) O145:H28 can cause severe disease in humans and is a predominant serotype in STEC O145 environmental isolates. Here, comparative genomics was applied to a set of clinical and environmental strains to systematically evaluate the pathogenicity potential in environmental strains. While the core genes-based tree separated all O145:H28 strains from the non O145:H28 reference strains, it failed to segregate environmental strains from the clinical. In contrast, the accessory genes-based tree placed all clinical strains in the same clade regardless of their genotypes or serotypes, apart from the environmental strains. Loss-of-function mutations were common in the virulence genes examined, with a high frequency in genes related to adherence, autotransporters, and the type three secretion system. Distinct differences in pathogenicity islands LEE, OI-122, and OI-57, the acid fitness island, and the tellurite resistance island were detected between the O145:H28 and reference strains. A great amount of genetic variation was detected in O145:H28, which was mainly attributed to deletions, insertions, and gene acquisition at several chromosomal “hot spots”. Our study demonstrated a distinct virulence gene repertoire among the STEC O145:H28 strains originating from the same geographical region and revealed unforeseen contributions of loss-of-function mutations to virulence evolution and genetic diversification in STEC.

## 1. Introduction

Shiga toxin-producing *Escherichia coli* (STEC) consists of a group of genetically and phenotypically diverse strains differing greatly in pathogenicity. While some strains can cause severe disease, others are only associated with mild diarrhea or no disease at all [1]. Such variation is attributed in part to the differences in their genetic makeup, especially the repertoire of virulence genes. The genome of STEC is extremely diverse, consisting of a core genome that is conserved among the STEC strains and an accessory genome containing strain, lineage, or pathovar-specific genes [2,3,4]. Enterohemorrhagic *E. coli* (EHEC), originally referring to STEC strains that cause hemolytic uremic syndrome (HUS) or hemorrhagic colitis (HC), expresses Shiga toxin (Stx) and enterohemolysin, and produces the attaching-and-effacing (A/E) lesions on intestinal epithelial cells [5]. Although STEC O157:H7, a prototype of EHEC, has been considered the most frequent cause of STEC-linked outbreaks, non-O157 strains are increasingly characterized as EHEC due to their association with HUS. Some of these strains, similar to O157:H7, are typical EHEC [6,7], whereas many are atypical EHEC, including the STEC O104:H4 linked to the 2011 large outbreak of hemorrhagic diarrhea in Europe [8]. These O104:H4 strains lack the locus of enterocyte effacement (LEE) harboring the genes responsible for formation of A/E lesions but carry two plasmids conferring cells’ aggregative capability and multi-drug resistance [9]. Therefore, the pathogenicity of a STEC strain is a combinational effect of the virulence genes it carries.

With more strains of the same species being sequenced, new genes are continually discovered. Examination of over 2000 *E. coli* genomes revealed a large pangenome (~89,000 total genes) and a small core genome (~3100 genes, conserved in all genomes) [10], which is attributed at least in part to its versatile lifestyle [11]. STEC cycles between host and nonhost environments. The accessory genome can be seen as an arsenal for STEC, one that makes them well equipped to adapt to different environments: from ruminants or transit hosts to the environment, and/or to humans. In fact, the accessory genome has been suggested to contain niche-specific information and could be used to differentiate the isolates’ origins [12,13,14,15]. Comparative genomics of STEC revealed a parallel evolutionary pathway: the acquisition and loss of virulence elements have occurred in parallel multiple times in separate lineages [6,7,16]. The contribution of horizontal gene transfer (HGT) to genome and virulence evolution in STEC is far more extensive than ever thought. Comparative genomic analysis of the EHEC strain EDL933 and *E. coli* K-12 strain MG1655 discovered over 1000 new genes in EDL933 [17]. These genes are often organized as clusters of diverse sizes known as genomic islands (GIs) or pathogenicity islands (PAIs) if they harbor virulence genes. GIs offer selective advantages to an organism and increase bacterial fitness, part of this benefit comes from the large number of genes that come with the acquisition of a new GI [18]. In STEC, besides virulence genes, genes related to stress resistances, autoaggregation, adherence, and biofilms are associated with GIs [19,20,21]. GIs are typically large, ranging from 10–200 Kb, and insert into the chromosome near tRNA genes with flanking direct repeats (DRs), and transposons or insertion sequences (ISs) are commonly found near the ends [22]. Thus, GIs represent unstable portions of the genome that are subject to excision, rearrangement, and transfer, and contribute to the rapid evolution of STEC [23,24].

STEC O145 is among the top six non-O157 serogroups that are most frequently associated with human disease in the U.S. [25,26,27]. Both serotypes O145:H28 and O145:H25 are capable of causing HUS, although outbreaks linked to O145:H28 are more frequent than O145:H25. Our previous work demonstrated that STEC O145:H28 shares a common evolutionary lineage with O157:H7 [6], but differs from O145:H25 in both the evolutionary path and traits associated with virulence and colonization [28]. Comparative phenotypic analyses of a clonal population of environmental STEC O145:H28 isolates carrying the same genotype as the clinical strains revealed that only a subset of cattle isolates exhibited comparable virulence traits with the clinical strains [29], supporting the report that only a minor subset of bovine isolates have the potential to cause disease in humans [12]. This divergence in STEC pathogenicity among the genotypically highly related strains could be explained at least in part by the differential expression of Shiga toxin, as demonstrated in STEC O145:H28 isolates [30]. However, in addition to prophages, many virulence genes in STEC are located on other Mobile Genetic Elements (MGEs) such as PAIs, Integrative and Conjugative Elements (ICEs), and plasmids. In fact, besides LEE and pEHEC, there are several GIs and PAIs reported in diverse STEC strains conferring pathogen cells advantages in various aspects of survivability in diverse ecological niches, including adherence, biofilm formation, and stress resistance [19,20,31]. In this study, we assembled a set of STEC O145:H28 environmental and clinical strains, all with high quality complete genome sequences, and used *E. coli* K-12 substrain MG-1655, the O157:H7 strain EDL933, and two O145:H25 strains as references. We first examined the conservation of 333 *E. coli* virulence genes deposited in the Virulence Factor DataBase [32,33], followed by comparative genomic analyses of ten PAIs/GIs known to contribute to pathogenicity and fitness traits in STEC to assess the pathogenicity potential in STEC environmental isolates.

## 2. Materials and Methods

### 2.1. Molecular Typing and Comparative Genomic Analyses

The bacterial strains, source/isolation year, and their genomic characteristics are listed in Table 1. The phylogroups were determined in silico using the Clermont method [34]. The Multi-Locus Sequence Typing (MLST)-based genotypes were determined using MLST 2.0 service at Center for Genomic Epidemiology via Warwick scheme [35] in April 2021. The pangenomes of STEC were calculated using Roary (V3.13.0), a pangenome bioinformatic pipeline as described previously [36]. Briefly, the GFF files of each genome were retrieved from GenBank in December 2020 and the Coding DNA Sequences (CDSs) were extracted from each input genome and converted to protein sequences. Homologous proteins were identified by all-against-all comparison with 95% as the minimum BLASTP percentage identity. The relatedness of the STEC strains was first evaluated by a core genome-based tree. All core genes were aligned using MAFFT (V7.471) followed by construction of a maximum-likelihood tree with 1000 bootstrap in IQ-TREE (version 2.1.2) using the best-fit model GTR + F + I, as selected by ModelFinder [37,38]. To identify the strains carrying more common genes, an accessory genome-based tree was constructed using a FASTA file of each genome with the binary of presence and absence of the accessory genes in Roary. An approximately maximum-likelihood tree was constructed in FastTree with the default settings (Jukes-Cantor + CAT model or GTR + CAT model) for generalized time-reversible (GTR) models.

### 2.2. Identification of Virulence Genes

The putative virulence genes were identified using VFanalyzer at The Virulence Factor DataBase (VFDB) using pre-annotated genomes in GenBank format. Each pre-annotated genome was uploaded to VFanalyzer to scan for all *Escherichia* Virulence Factors (VFs) deposited in VFDB as of April 2021, which included genes identified in 37 *E. coli* strains belonging to adherent invasive *E. coli* (AIEC), avian pathogenic *E. coli* (APEC), enteroaggregative *E. coli* (EAEC), Shiga toxin-producing enteroaggregative *E. coli* (StxEAEC), enterohemorrhagic *E. coli* (EHEC), enteropathogenic *E. coli* (EPEC), enterotoxigenic *E. coli* (ETEC), neonatal meningitis-associated *E. coli* (NMEC), and uropathogenic *E. coli* (UPEC). The identified virulence genes were further verified manually by BLASTn search of a database containing all STEC strains in Geneious Prime with a threshold of 80% for gene coverage and 75% for sequence identity.

### 2.3. Detection of GIs and PAIs

GIs were first revealed using IslandViewer 4 [46] with the genome of *E. coli* K-12 strain MG1655 as a reference. A GI was called when a prediction was made by at least one of the three methods (IslandPath-DIMOB, SIGI-HMM, and IslandPick). The GIs and PAIs known to contribute to stress resistance and/or pathogenicity in enteric pathogens (Table 2) were further examined by performing BLAST searches against a custom database containing all genomes described in this study. When a complete GI or PAI was not detected, all CDSs encoded by the query GI or PAI were used to search the genome of the testing strain by BLASTP to reveal if any homologs were present in the testing strain. Presence of key virulence genes on each PAI were revealed by BLASTn of each virulence gene followed by mapping their chromosomal locations in Geneious Prime. Strain EDL933 harbors two tellurite resistance islands (TRIs) (OI-43 and OI-48), exhibiting over 99% identity. The relatedness of the GIs or PAIs was assessed by multiple sequence alignment using Clustal Omega in Geneious Prime followed by construction of a consensus tree using the Neighbor-Joining method and Jukes-Cantor for the genetic distance model.

### 2.4. Statistical Analyses

The difference in gene content between the clinical and environmental strains was assessed by an unpaired *t*-test with a 95% confidence interval using Prism 8.0 (GraphPad Software). The association of virulence genes with isolation origin was evaluated with Fisher’s exact test with Bonferroni correction for multiple comparisons using R version 4.1.0.

## 3. Results

### 3.1. Comparative Genomics of STEC O145:H28

Comparative genomic analyses of the 19 STEC O145:H28 strains revealed a total of 8527 genes, among which, 3702 genes were detected in all genomes (core genes, 43.4%) (Figure 1A). The accessory genes included 2229 genes present in more than three, but less than 19 genomes (shell genes) and 2596 genes present in less than three genomes (cloud genes), implying that STEC O145:H28 strains carry a large number of dispensable genes. As expected, the pangenome size (total genes) increased greatly when the three non O145:H28 STEC reference genomes were included in the analyses, as well as the number of cloud genes (Appendix A). Among the 10,288 genes detected in all STEC genomes, 29.1% (2998) were core genes and 70.9% were accessory genes, supporting that STEC possesses a large accessory genome. Examining the strain-specific genes revealed that the clinical O145:H28 strains carried more strain-specific genes than the environmental strains (unpaired t test, *p* = 0.0095) (Figure 1B). The Belgium outbreak strain RM13516 carried the greatest number of strain-specific genes, followed by the U.S. outbreak strain RM13514. Among the environmental strains, the cattle isolates RM9872-C1 and RM9154-C1 carried more strain-specific genes than any other strains. The majority of the strain-specific genes in O145:H28 were MGE-associated and hypothetical genes. Others included genes related to transporters, toxin-antitoxin systems, secretion systems, the type three secretion system (T3SS) effectors, and the type IV pilus.

### 3.2. Strain Relatedness

The relatedness of the STEC strains was examined by core and accessory genes-based trees. When the 2998 core genes were used for a similarity calculation, all STEC O145:H28 strains were placed in a large clade, apart from the cluster containing the O145:H25 strains and the O157:H7 strain EDL933 (Figure 2A). Within the O145:H28 clade, the Belgium outbreak strain RM13516 was placed in the same cluster as the Japanese clinical isolate 112648, the U.S. clinical isolate 95-3192, and all the *stx*_1a_-positive environmental isolates, whereas the U.S. outbreak strain RM13514 was placed in a different cluster with the Japanese clinical isolates 122715 and 10942, the U.S. clinical isolate 2015C-3125, and all the *stx*_2a_-positive environmental isolates (Figure 2A).

A distinct topology was observed when the accessory genes were used to assess strain relatedness. Regardless of the serotype or genotype, all clinical strains were placed in the same clade, where three Japanese strains formed a cluster, the two U.S. strains 95-3192 and 2015C-3125 also grouped together, and the outbreak strains RM13514 and RM13516 were placed in the same cluster as the two O145:H25 strains (Figure 2B). All environmental isolates except RM9154-C1 were placed in a separate clade, consisting of two clusters, one containing all the *stx*_1a_-positive environmental isolates and the other one containing all the *stx*_2a_-positive environmental isolates. The environmental isolate RM9154-C1 formed a singleton and appeared to share more accessory genes with the clinical strains than with the other environmental strains (Figure 2B).

### 3.3. Conservation of E. coli Virulence Genes in Environmental Strains

The 333 virulence genes retrieved from the VFDB encode VFs belonging to 10 major functional categories and contribute to *E. coli* pathogenicity in diverse pathotypes (Figure 3A). Of the 333 genes examined, 165 were detected in EDL933, and 140 and 132 were detected in O145:H25 strains CFSAN004176 and CFSAN004177, respectively (Figure 3B). Among the O145:H28 strains, the number of virulence genes ranged from 126 in the environmental strain RM12275-C1 to 151 in the clinical strain 112648. This variation was mainly due to the difference in number of genes encoding non-LEE encoded T3SS effectors and genes related to adherence and iron uptake (Appendix A). There was no significant difference in the number of total virulence genes between the clinical and the environmental O145:H28 strains (unpaired *t* test, *p* > 0.05), nor in the number of virulence genes in each VF category examined (unpaired t test, *p* > 0.05). Furthermore, none of the detected virulence genes were specifically associated with the isolation source (Fisher’s test, adjust *p* > 0.05).

The genes encoding curli fimbriae (*csgDEFG* and *csgBAC*), *E. coli* common pilus (*ecpRABCDE*), *E. coli* laminin-binding fimbriae (*elfADCG*), hemorrhagic *E. coli* pilus (*hcpCBA*), and type I fimbriae (*fimBEAICDFGH*) were detected in all strains; however, loss-of-function mutations were common, including a nonsense mutation in *csgB* and point deletions in *ecpE*, *elfC*, *elfD*, *elfG*, *fimB*, and *fimD* in multiple strains (Appendix A). Similarly, the genes encoding protein adhesins intimin, intimin-like adhesin EaeH, porcine attaching-effacing associated protein Paa, and ToxB were detected in most strains, although point deletions were detected in *eae*, *eaeH*, and *toxB* (Appendix A). Several virulence genes were only detected in the O145:H25 strains, including the *cfaABCD* genes, encoding the colonization factor antigen I (CFA/I) fimbriae, and homologs of P fimbriae genes *papC* and *papD* (Appendix A).

Seven out of the 20 autotransporters genes, *ag43*, *cah*, *ehaA*, *ehaB*, *espI*, *espP*, and *upaG*, were detected in the majority of STEC strains examined. The *ag43* encodes a 1039-aa autotransporter adhesin Ag43, exhibiting 69.0% identity with Cah, a 949-aa calcium-binding autotransporter protein. Strain EDL933 carried two copies of *cah* but no *ag43*, whereas all O145:H28 strains carried both *ag43* and *cah* except for strains 10942 and RM12275-C1. Neither *ag43* nor *cah* was detected in the O145:H25 strains. Loss-of-function mutations were detected in both *ag43* and *cah*, including point insertions, point deletions, transposon insertions, and large deletions (Appendix A). The serine protease autotransporter EspI exhibits 48.8% identity with the serine protease autotransporter EspP. The *espI* was detected only in the U.S. outbreak strain RM13514 and the two STEC O145:H25 strains, whereas the *espP* was detected in strain EDL933 and the majority of O145:H28 strains including RM13514. A point deletion in *espP* was detected in RM12275-C1 (Appendix A). The *upaG* gene encodes a 1447-aa YadA-like family protein that mediates aggregation, biofilm formation, and adhesion to extracellular matrix proteins. A homolog of *upaG* was identified in all strains, ranging in size from 4620 bp in RM13514, 4767 bp in EDL933, 4851 bp in the two O145:H25 strains, to 5130 bp in other O145:H28 strains. Two point insertions in *upaG* were identified in strain 10942 (Appendix A). The *ehaA* and *ehaB* encode a 1327-aa and a 980-aa autotransporter adhesin, respectively, contributing to attachment and biofilm formation [50,51]. Both *ehaA* and *ehaB* were present in all strains including the two O145:H25 strains; however, IS insertions in both genes and a point deletion in *ehaA* were detected in multiple strains (Appendix A).

Four genes (*ibeA*, *ibeB*, *ibeC*, and *tia*) related to *E. coli* invasion were examined. While no homologs of *ibeA* or *tia* were detected in any of the strains examined, homologs of *ibeB* and *ibeC* were detected. For *ibeC*, in addition to a highly similar homolog (%Identity, 95.3), a distantly related homolog (% Identity, 78.0) was detected on the plasmid pEHEC in all O145:H28 strains and in EDL933, although a point deletion was present in the plasmid-borne *ibeC* in strain 2015C-3125 (Appendix A).

Among the 33 genes related to iron uptake, homologs of 13 genes were detected. The genes encoding heme utilization system (*chuAS* and *chuTWXYU*) were present in all O145:H28 strains. In contrast, none of the heme utilization genes were detected in O145:H25. The genes involved in aerobactin siderophore synthesis (*iucABCD*) and the ferric siderophore receptor gene *iutA* were detected in all O145:H28 strains except RM9154-C1. A homolog of *iroB*, encoding the salmochelin biosynthesis C-glycosyltransferase, was present in all O145:H28 strain. IS insertions and point deletions in *iucC*, *iucD*, and *iroB* were detected (Appendix A). Strain EDL933 carried two copies of *iroB*; one was intact while the other contained a point deletion, similar to the *iroB* in strain 2015C-3125.

A total of 34 genes involved in biosynthesis of the T3SS apparatus was detected in all O145 strains, although point deletions were present in genes *cesL*, *escL*, *escN*, *escR*, *escU*, and *glrR*. Among the seven LEE-encoded T3SS effector genes, point deletions in *espF*, *espG*, *espH*, and *tir*, and a large deletion in *espF* were detected (Appendix A). The number of non-LEE-encoded T3SS effector genes varied greatly among the strains, from 51 in EDL933 to 29 in O145:H28 strain RM12275-C1. This difference was largely attributed to the variation in possession of genes encoding EspL2, EspN, EspX4, EspX6, EspX7, LifA, NleB2-1, NleC, NleG7, NleH1-2, and TccP2 (Appendix A) and of loss-of-function mutations in *espN*, *espV*, *espW*, *espX1*, *espX4*, *espX7*, *espY4*, *espY5*, *nelC*, and *nelG2-3* in various strains (Appendix A).

There are three type six secretion system (T6SS) gene clusters reported in *E. coli* [52], among them, the T6SS-1 and T6SS-2 are the most common. Among the 22 genes related to T6SS-2, homologs of 15, eight, and five genes were identified in strain EDL933, both O145:H25 strains, and the majority of O145:H28 strains, respectively (Appendix A). No homologs of the genes related to T6SS-1 or T6SS-3 were identified in any of the O145:H28 strains examined.

The toxin genes detected in STEC included plasmid-borne enterohemolysin genes *hlyCABD*, chromosome-borne hemolysin gene *hlyE*, and Shiga toxin genes *stx*. All strains except strain 95-3192 carried *hlyCABD*, although point deletions in *hlyB* and *hlyC* were detected in multiple strains (Appendix A).

### 3.4. Diversification of Pathogenicity Islands (PAIs) in STEC

Eight PAIs contributing to the pathogenicity of enteric pathogens and two GIs contributing to *E. coli* stress resistance were further examined (Table 2). All strains carry a complete LEE; however, variations in PAIs OI-122 and OI-57, TRI, locus of proteolysis activity (LPA), and acid fitness island (AFI) were detected. Additionally, although a complete locus of adhesion and autoaggregation (LAA) island was not detected in any of the strains examined, LAA modules were identified.

#### 3.4.1. LEE

The LEE in strain EDL933 corresponds to the O-island 148 (OI-148) and inserts downstream of tRNA gene *selC* (Table 3). A complete LEE was identified in all O145:H28 strains, ranging in size (bp) from 46,601 to 47,785. Similar to the EDL933 LEE, the O145:H28 LEEs mainly harbor genes related to biosynthesis of T3SS apparatus and the genes encoding T3SS effectors. Furthermore, all the O145:H28 LEEs insert downstream of *selC* and encode γ subtype intimins (Table 3). In contrast, the O145:H25 LEEs insert downstream of the tRNA gene *pheV* and encode β subtype intimins (Table 3). The O145:H25 LEEs are much larger in size and form a separate cluster compared with the O145:H28 LEEs (Table 3 and Appendix A). Besides the genes related to biosynthesis of T3SS apparatus and T3SS effectors, the O145:H25 LEEs harbor additional genes encoding MGIs, transporters, and transcriptional regulators. 

#### 3.4.2. OI-122

The OI-122 in EDL933 is 23,455 bp and inserts downstream of *pheV*. A complete or partial OI-122 was identified in all O145:H28 strains except RM12275-C1, ranging in size (bp) from 12,900 to 53,262 (Table 3). In RM12275-C1, the integration site for OI-122 is unoccupied. A sequence alignment of OI-122 failed to separate the clinical O145:H28 strains from the environmental isolates (Appendix A).

The complete OI-122 contains three modules, each bordered by MGIs. Module I carries virulence gene *pagC*, encoding a membrane protein involved in serum resistance in *Salmonella enterica*. A complete module I was identified only in strain RM13516, exhibiting 99.6% identity with the OI-122 module I in EDL933. Module II carries genes encoding T3SS effectors EspL, NleB, and NleE. A complete module II was present on all OI-122s identified. Both O145:H25 strains carried a second copy of module II, located on the LEEs. Module III encodes an Efa1/LifA-like adherence protein. A full-length *efa1* gene was detected only in strains RM13516 and CFSAN004177. A truncated *efa1* was detected in strain CFSAN004176 and in 10 other O145:H28 strains. Module III was detected in five out of seven O145:H28 clinical strains and in six out of 12 environmental O145:H28 strains. In both O145:H25 strains, a second copy of module III was present on the LEEs, similar to the module II. In fact, the 17 kb DNA segment containing both module II and III on the LEE was nearly identical to the corresponding region on OI-122.

#### 3.4.3. OI-57

The OI-57 in EDL933 belongs to the genome of prophage CP-933O, carrying virulence genes *adfO* (*paa*) and *ckf*, and is associated with the highly pathogenic seropathotypes A and B [53]. A homolog of OI-57, ranging in size (bp) from 14,385 to 137,948 was identified in all O145:H28 strains, but not in either of the O145:H25 strains (Table 3). Similar to EDL933 OI-57, the O145:H28 OI-57s insert in gene *ompW*. Both *adfO* and *ckf* were present on O145:H28 OI-57s but annotated as *paa* and the type I toxin-antitoxin system Hok family toxin gene, respectively. Sequence alignment of OI-57 failed to separate the clinical O145:H28 strains from the environmental strains (Appendix A).

#### 3.4.4. TRI

Strain EDL933 harbors two TRIs, one located downstream of tRNA gene *serW* (OI-43) and the other inserted downstream of tRNA gene *serX* (OI-48) (Table 4). Both islands are bordered by a 14 bp DR sequence (Figure 4A). Spontaneous deletions of TRIs in strain EDL933 occur via recombination between the DRs flanking each TRI [54]. In both O145:H25 strains, a TRI was identified, located downstream of tRNA gene *ileX* (Table 4). Among the O145:H28 strains examined, only RM13514 harbors two TRIs, one inserted downstream of *serX* (TRI-1), similar to the OI-48 in strain EDL933, and the other inserted downstream of *ileX* (TRI-2), just as the TRIs in O145:H25 strains (Figure 4B). All other O145:H28 strains harbor one TRI, such as the OI-48 in strain EDL933, inserted downstream of *serX*. The two 14 bp DRs bordering the OI-43 or OI-48 in EDL933 were present in the Belgium outbreak strain RM13516 (Figure 4C). In other O145:H28 strains including RM13514, only one 14 bp DR was detected. Two 12 bp DRs bordering the TRI-2 in strain RM13514 were identified, while in both O145:H25 strains, only one 12 bp DR was detected (Figure 4D). A sequence alignment of TRIs grouped the O145:H25 TRIs with the RM13514 TRI-2, apart from all other TRIs, consistent with the TRI chromosomal integration sites (Appendix A).

#### 3.4.5. LAA

LAA is a large PAI initially identified in STEC O91:H21 strain B2F1 and present mainly in LEE-negative STEC strains [20]. LAA contains four modules, each flanked by ISs and/or DRs. The main virulence genes on the 11,465 bp module I are *sisA* and *hes*, related to attenuation of host immune response and adherence to epithelial cells, respectively. None of the strains examined harbored a complete module I, although *sisA* was detected in all O145:H28 strains and located on the TRIs (TRI-1 in strain RM13514). Two copies of *sisA* were identified in EDL933, located on OI-43 and OI-48, respectively. Neither *sisA* nor *hes* was detected in the O145:H25 strains.

The main virulence genes on the 21,363 bp module II are *iha* and *lesP*, encoding adhesin Iha and a serine protease autotransporter, respectively. A fragment of module II was identified in all O145:H28 and O145:H25 strains. Two copies of the same module II fragments were identified in EDL933. Similar to *sisA*, this partial module II was located on the TRIs in O145:H28 strains and in EDL933. In fact, *sisA* inserts in the partial LAA module II. Unlike module II in strain B2F1, the partial module II in EDL933 and in all O145:H28 strains, except RM13514, carried *iha* but not *lesP*. RM13514 carried both *iha* and *lesP*, located on TRI-1 and TRI-2, respectively. No *iha* was detected in O145:H25 strains, although two copies of *lesP* were identified, located on the TRI and the pEHEC plasmid, respectively.

The main virulence genes on the 25,932 bp module III are *pagC*, related to serum resistance in *Salmonella enterica*, *tpsA* and *tpsB*, encoding a two-partner secretion system. A partial module III was detected in all O145:H28 strains and in strain EDL933, similarly to the partial module II, located on TRIs (TRI-1 in RM13514). No homologs of module III genes were identified in O145:H25 strains. The *pagC* was only detected in strains RM13514, RM13516, and EDL933, but located on the OI-122. The *tpsA* gene was not detected in any of the strains, whereas a 539 bp DNA fragment exhibiting 98.7% identity with the *tpsB* gene was detected in all O145:H28 strains and in strain EDL933, but not in the O145:H25 strains.

The main virulence gene on the 21,088 bp module IV is *ag43*, encoding an adhesin that promotes autoaggregation. A partial module IV ranging in size from 14,084 bp to 15,593 bp was identified in all O145:H28 strains. This partial module was also located on the TRI (TRI-1 in RM13514), about 13 Kb apart from the partial module III. The gene *cah* was detected on this partial module IV. A second partial module IV, located on the OI-122, was detected in all O145:H28 strains except 10942, RM9154-C1, and RM12275-C1, ranging in size from 14 kb to 19 kb (Table 4). A third partial module IV was detected in outbreak strain RM13516, located on the integrative element 06 (IE06). The *ag43*, but not *cah*, was present on the partial module IV located on OI-122 and IE06, respectively (Table 4). In strain EDL933, a partial module IV was present on OI-43 and OI-48, but not on OI-122. A fragment of module IV was detected in both O145:H25 strains, which, similarly to the LAA module II fragment, was located on the TRI (Table 4).

#### 3.4.6. Others

The AFI in STEC O145:H25 strains is highly similar to the AFI in *E. coli* K-12 strain MG1655, carrying genes related to acid resistance (*yhiF*, *yhiD*, *hdeBAD*, *gadE*, *gadW*, *gadX*, and *gadA*) and genes encoding the efflux pump (*mdtEF*). The AFI in all O145:H28 strains is highly similar to the AFI in strain EDL933, besides the acid resistance genes it also harbors the iron acquisition genes *chuSA* and *chuTWXYU*. Therefore, the AFIs in all O145:H28 strains are larger in size (~22 Kb) than the AFIs in both O145:H25 strains and in strain MG1655 (~13 Kb) (Table 4).

The 37 kb LPA island often inserts downstream of *selC*; therefore, it is likely present in LEE-negative strains. No homologs of LPA genes were identified in any of the O145:H28 strains examined; however, a 22 kb GI, located immediately downstream of *selC*, was identified in both O145:H25 strains. This GI exhibited high sequence similarity with the LPA only at the two border regions. Neither *espP* nor *iha* was detected on this GI. No homologs of the high-pathogenicity island (HPI), subtilase encoding pathogenicity island (SE-PAI), or locus of heat resistance (LHR) were identified in any of the strains examined (Table 2).

## 4. Discussion

The vast genetic variation in STEC poses challenges in detection of hypervirulent strains in diverse environmental and food samples. *E. coli* holds an open pangenome and evolves more rapidly compared to species with closed pangenomes. Emergence of hypervirulent strains could occur in any niches when the right set of virulence genes become available and assembled successfully. Such events are likely recorded in the accessory genome. The clear separation between clinical and environmental strains revealed in our study supports this speculation and suggests that accessory genes indeed carry information reflecting the recent strain evolution history (short evolution). This speculation is further supported by the clear separation of the *stx*_1a_-positive O145:H28 isolates from the *stx*_2a_-positive O145:H28 isolates. Formation of the singleton by the cattle isolate RM9154-C1 is consistent with our recent report that strain RM9154-C1 harbors a *stx*_2a_-prophage differing largely from the *stx*_2a_-prophages in other environmental strains [30]. In fact, RM9154-C1 carries a *stx*_2a_-prophage highly similar to the *stx*_2a_-prophages in several hypervirulent strains including the O145:H28 strain RM13514, O157:H7 strain EDL933, and O104:H4 strain 2011C-3493 [30]. Unlike the core genome, which generally encodes essential functions, the accessory genome often encodes functions to provide additional traits for bacteria to expand their ecological niches [11,55]. The virulence genes repertoire of each STEC strain is likely impacted by the microbial community it resides in, as well as stress factors it encounters, considering that many common stresses promote HGT. A recent study revealed that clinically important STEC strains have emerged in multiple sublineages of bovine-adapted linage, implying that certain ecological niches, such as the bovine intestinal environment, may have provided selection pressures for STEC/EHEC virulence genes [14].

Our study revealed widespread loss-of-function mutations in diverse virulence genes. Loss-of-function mutations are an important mechanism in bacterial niche adaptation, including in the infected hosts [56,57]. Mutation in a regulatory gene could rewire the cell metabolism, providing the bacteria a largely altered physiology to exploit a resource in a new niche, as demonstrated for *rpoS* and *rcsB* mutations in STEC [58,59]. A mutation in *fimH* was linked to increased bacterial uroepithelial adhesion and bladder colonization [60], while loss of curli was suggested to improve pathogens’ survival following infections [61,62,63]. Similarly, mutations in *cah* were suggested to confer STEC a selective trait when autoaggregative properties become detrimental [64], and mutations in autotransporter gene *ompF* were suggested to provide protection to *E. coli* cells from toxic compounds, such as tetracycline [65]. Our study revealed mutations in additional loci leading to silencing the production of curli, common pili, laminin-binding fimbriae, type I fimbriae, adhesin proteins, and autotransporters in STEC, although further studies are needed to understand the selection pressures for these mutations as well as the ecological benefits in maintaining these mutations in the STEC population.

Many bacterial pathogens utilize T3SS to inject toxins and/or effectors into host cells. These toxins or effectors have evolved to counteract the host’s defenses via modulating host cell signaling pathways [66,67]. Our study revealed widespread mutations in genes related to T3SS apparatus and effectors, including Tir, one of the best characterized T3SS effectors in STEC. Tir functions as a receptor for the outer membrane protein intimin, mediates direct interaction between bacterial pathogen cells and the eukaryotic host cells and recruits α-actinin to the pedestal, a critical step in formation of A/E lesions [68]. Interestingly, loss-of-function mutations were also detected in *eae*. Therefore, T3SS genes are likely under positive selection in STEC and undergo adaptive changes, similarly to the genes encoding surface proteins and fimbrial/non-fimbrial surface structures that mediate the interactions with other bacteria, phages, and/or host immune systems [69,70].

Our study provides genomic insights into the divergent evolution of virulence in STEC. Consistent with a previous report that O145:H25 differs from O145:H28 phylogenetically and carries more fimbrial genes than O145:H28 [28], the CFA/I fimbriae genes were only detected in O145:H25. CFA/I fimbriae mediate the colonization of ETEC cells to human intestine [71]. This gene cluster is also present in a large number of STEC strains belonging to phylogroup B1, including the STEC O104:H4 outbreak strain 2011C-3493. However, compared to O145:H28, O145:H25 lacks the iron uptake genes on the AFI (heme utilization system) and the TRI (aerobacterin siderophore and ferric siderophore receptor), implying distinct physiological trait and fitness between the two serotypes, especially under iron-limiting conditions and in infected humans. Additionally, O145:H25 lacks other virulence genes located on the TRI, and the entire pathogenicity island OI-57, but carries a much larger LEE compared with O145:H28, which is likely a result of the transposon-mediated insertion of a 17 kb DNA fragment duplicated from OI-122.

Great genetic variations were also detected among the O145:H28 strains. Among the 333 virulence genes examined, the number of detected genes ranges from 128 to 154. This difference is mainly attributed to the differences in the number of genes encoding non-LEE-encoded T3SS effectors, and the genes encoding autotransporters and toxins. Since environmental isolate RM12275-C1 lacks OI-122, it lacks all OI-122-encoded virulence genes. The OI-122 in O145:H28 varies greatly in size and the number of genes encoding MGEs, suggesting that this chromosomal region is a “hot spot” for insertions, deletions, and/or recombination. The gene *ompW* serves as another “hot spot” in the O145:H28 genome since it contains the integration site for OI-57. In EDL933, OI-57 is part of the prophage CP-933O genome. In O145:H28 strains, the OI-57s are overlapped with the prophages and integrated within the *ompW*, including the *stx*_2a_-prophage in RM12367-C1 [30]. Therefore, the variation in OI-57 is likely due to the difference in genomes of the prophages inserted in *ompW*.

Genetic variation in STEC O145:H28 is further contributed by the mobility of GIs. Strain RM13514 is the only one carrying two TRIs, one (TRI-1) highly similar to the EDL933 TRIs, whereas the other (TRI-2) highly similar to O145:H25 TRIs. TRI-1 appears to be fixed as one of the two 14 bp DRs is eliminated, but TRI-2 likely retains the mobility since the two 12 bp DRs flanking the TRI-2 are intact. The TRI in the Belgium outbreak strain RM13516 is the only one likely to retain mobility. Variation in O145:H28 TRIs are mainly attributed to the number of transposase genes and the genes involved in iron acquisition (*iucABCD* and *iutA*). The TRI in RM9154-C1 is much smaller than the TRIs in other strains due to lack of genes *iucABCD* and *iutA*. Great sequence diversity was also detected for the LAA modules. Consistent with a previous report that the LAA is largely present in LEE-negative strains, none of the strains examined carries a complete LAA. However, several key genes located on the four LAA modules were detected but were incorporated in TRIs. Duplications of the LAA module IV were detected in multiple strains and all are associated with transposons. Because the main virulence factor encoded by the LAA module IV is Ag43, variation in the LAA module IV would likely result in the difference in production of Ag43. Presence of the LAA modules on the TRIs in EDL933 perfectly explains the loss of bacterial adherence to human intestinal epithelial cells when the TRIs were excised from chromosome [54]. However, this impact would be averted if the LAA modules are located elsewhere, such as on the OI-122 as detected in this study.

## 5. Conclusions

The plasticity of the STEC genome provides the pathogen great potential for genome expansion and niche adaptation. Our study demonstrated the impacts of three major forces, HGT, loss-of-function mutations, and gene duplications, in driving divergent evolution of the genome and virulence of STEC. The high mutation rate in genes encoding protein adhesins, fimbriae, pili, autotransporters, and T3SS effectors implies the presence of positive selection for these genes in diverse ecological niches. Furthermore, environmental stresses that promote HGT or induce mutations accelerate genome and virulence evolution in STEC, including UV radiation, antimicrobial washes, and host defenses. Understanding the interactions between STEC and their indigenous community members would shed light on factors driving the emergence of STEC variants. Such knowledge is much needed for the development of effective mitigation strategies for detecting and controlling the transmission of STEC.

## Figures and Tables

**Figure 1 microorganisms-10-00866-f001:**
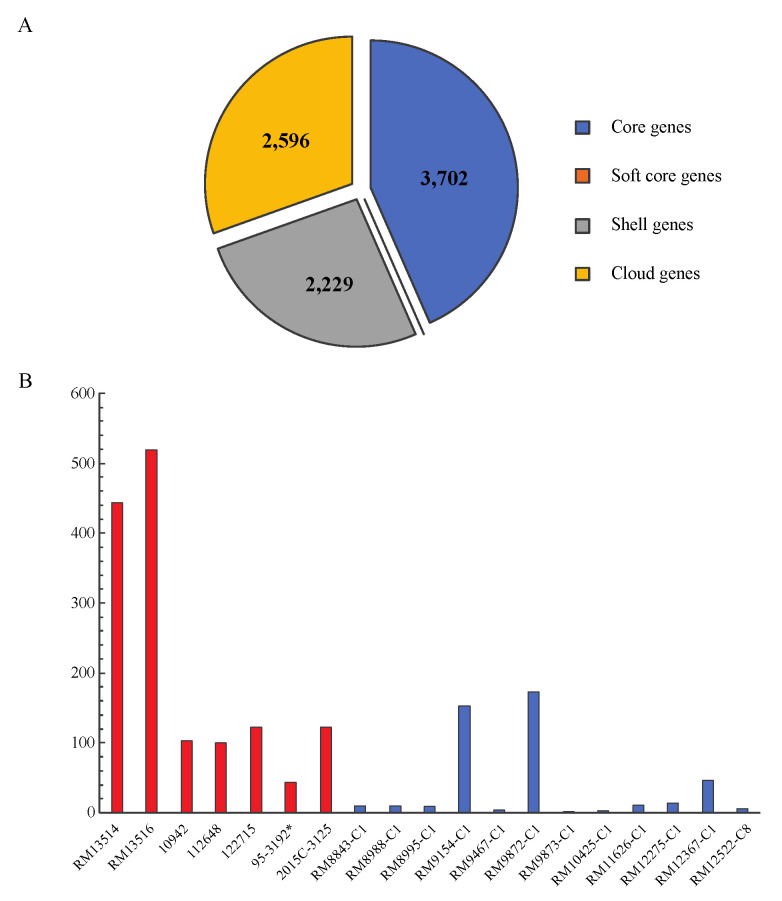
Comparative genomic analyses of STEC O145:H28. (**A**): Core and accessory genes of STEC O145:H28. The numbers of core and accessory genes were calculated in Roary as detailed in the Materials and Methods section. The core genes refer to all genes shared by all input genomes (n = 19); the soft-core genes refer to all genes present in any 18 input genomes (n = 18); the shell genes refer to all genes present in at least three but less than 18 input genomes (3 ≤ n < 18); the cloud genes refer to all genes present in less than three input genomes (n < 3). (**B**): Number of strain specific genes in STEC O145:H28. The number of strain specific genes was calculated in Roary as detailed in the Materials and Methods section. Red bars represent the clinical isolates whereas blue bars represent environmental isolates. * indicates that no plasmids were present in the published genome (GenBank Accession numbers are presented in Table 1).

**Figure 2 microorganisms-10-00866-f002:**
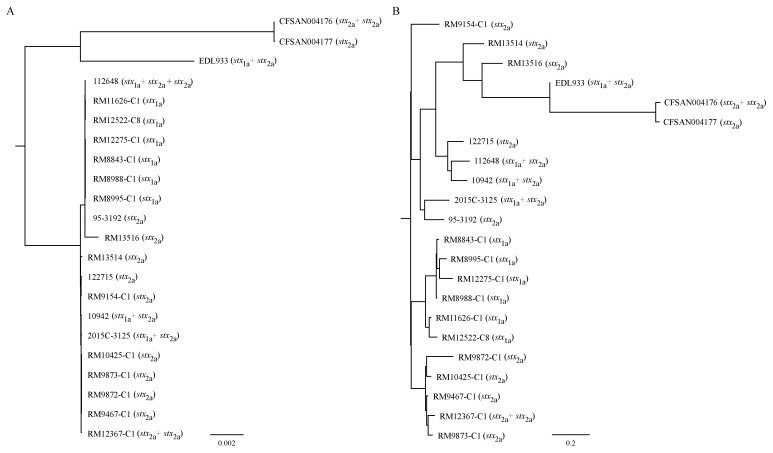
Relatedness of the STEC strains. (**A**): Core-genes based phylogenetic tree. The maximum likelihood-based phylogenetic tree was constructed based on the alignment of concatenated nucleotide sequences of 2998 homologous CDSs from each of the strains and using iqtree2 with the best fit model GTR + F + I as selected by ModelFinder and was assessed by bootstrapping with 1000 pseudoreplicates. (**B**): Accessory genes-based relatedness tree. The accessory genes of the STEC genomes were used to construct a FASTA file with the binary of presence and absence in Roary, followed by the construction of an approximately maximum-likelihood tree using FastTree as detailed in the Materials and Methods section. Thus, strains within the same cluster share more common accessory genes than the strains in a different cluster.

**Figure 3 microorganisms-10-00866-f003:**
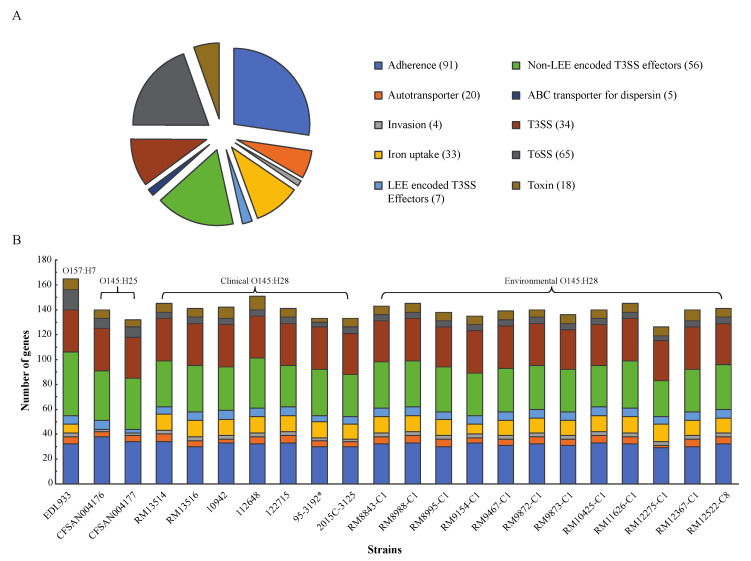
Detection of the *E. coli* virulence genes in STEC. (**A**): The functional categories of the *E. coli* virulence genes according to the classification in the VFDB [33]. A total of 333 virulence genes that contribute to the pathogenicity in various *E. coli* pathotypes were downloaded from the VFDB and grouped into 10 functional categories as detailed in the Materials and Methods. The list of genes and their association with the *E. coli* pathotypes are presented in Appendix A. (**B**): The number of *E. coli* virulence genes and their functional categories detected in each STEC strain. Presence of each virulence gene was verified by BLASTn search of a database containing all STEC strains in Geneious with a threshold of 80% for coverage and 75% for sequence identity. For genes carrying a loss-of-function mutation, they are marked as absent. * indicates that no plasmids were present in the published genome (GenBank Accession numbers are presented in Table 1).

**Figure 4 microorganisms-10-00866-f004:**
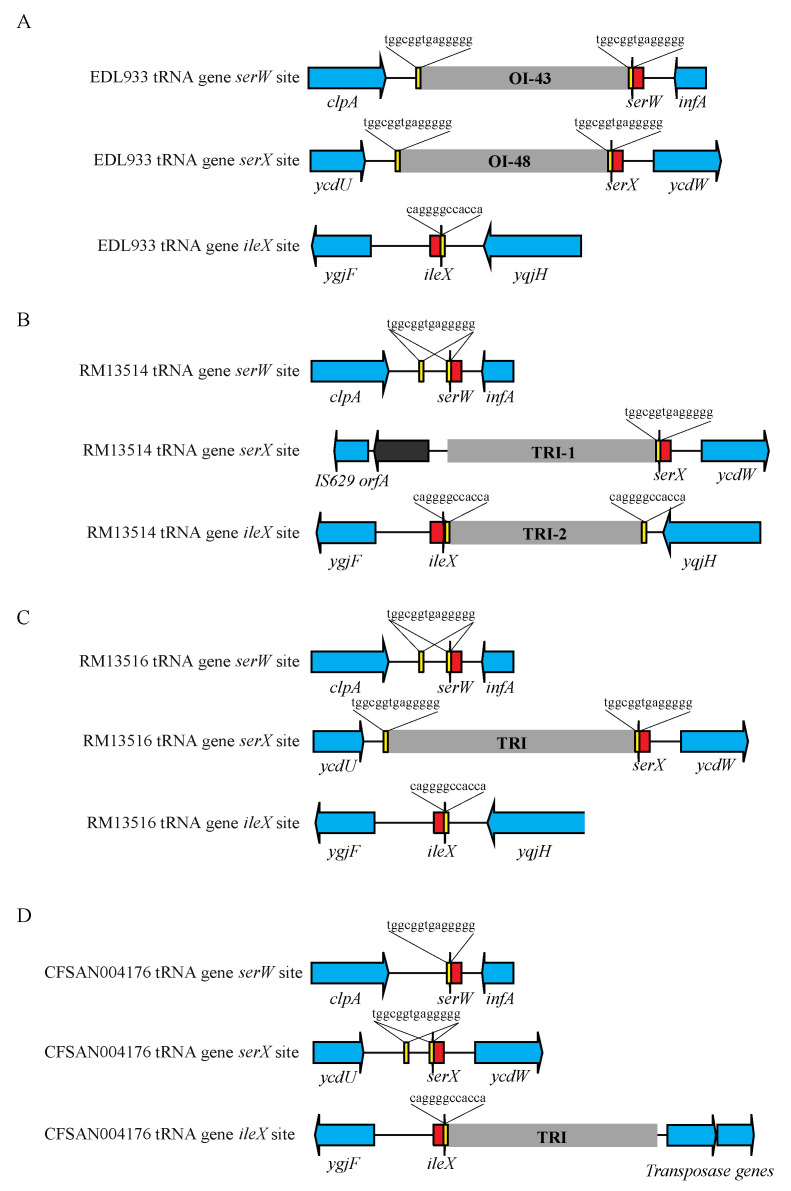
Sequence analyses of TRI insertion sites in STEC. (**A**): Sequence analyses of TRI insertion sites in STEC O157:H7 strain EDL933. Yellow blocks refer to the DRs bordering each TRI. The insertion site in tRNA gene *ileX* in strain EDL933 is unoccupied. (**B**): Sequence analyses of TRI insertion sites in STEC O145:H28 strain RM13514. Yellow blocks refer to the DRs bordering each TRI. The insertion site in tRNA gene *serW* in strain RM13514 is unoccupied. (**C**): Sequence analyses of TRI insertion sites in STEC O145:H28 strain RM13516. Yellow blocks refer to the DRs bordering each TRI. The insertion sites in tRNA genes *serW* and *ileX* in strain RM13516 are unoccupied. (**D**): Sequence analyses of TRI insertion sites in STEC O145:H25 strain CFSAN004176. Yellow blocks refer to the DRs bordering each TRI. The insertion sites in tRNA genes *serW* and *serX* in strain CFSAN004176 are unoccupied.

**Table 1 microorganisms-10-00866-t001:** Genomes used in this study.

Strains	^a^ Sources (Location, Year)	^b^ Serotype	Phylogroup/Genotype	*stx* Genes	Chromosome (bp)/ GenBank Accession #	^c^ Plasmids (bp)/GenBank Accession		References
						pEHEC	Others	
MG1655	Stool (USA, 1922)	O16:H48	A/ST10	N/A	4,641,652/U00096.3	N/A	N/A	[39]
EDL933	Ground beef (USA, 1982)	O157:H7	E/ST11	*stx*_1a_ + *stx*_2a_	5,528,445/AE005174.2	92,077/AF074613.1	N/A	[17,40]
CFSAN004176	Clinical (USA, 2003)	O145:H25	B1/ST5309	*stx*_2a_ + *stx*_2a_	5,193,734/CP014583.1	52,297/CP012493.1	95,721/CP012491.1; 34,714/CP012492.1	[28]
CFSAN004177	Clinical (USA, 2004)	O145:H25	B1/ST5309	*stx* _2a_	5,191,331/CP014670.1	52,297/CP012495.1	96,228/CP012494.1; 34,714/CP012496.1	[28]
RM13514	Clinical (USA, 2010)	O145:H28	E/ST32	*stx* _2a_	5,585,613/CP006027.1	87,120/CP006028.1	64,561/CP006029.1	[6]
RM13516	Clinical (Belgium, 2007)	O145:H28	E/ST6130	*stx* _2a_	5,402,276/CP006262.1	98,066/CP006263.1	58,666/CP006264.1	[6]
10942	Clinical (Japan, 2011)	O145:H28	E/ST32	*stx*_1a_ + *stx*_2a_	5,374,674/AP019703.1	92,337/AP019704.1	71,161/AP019705.1	[41]
112648	Clinical (Japan, 2011)	O145:H28	E/ST32	*stx*_1a_ + *stx*_2a_ + *stx*_2a_	5,488,534/AP019706.1	91,036/AP019707.1	N/A	[41]
122715	Clinical (Japan, 2012)	O145:H28	E/ST32	*stx* _2a_	5,418,961/AP019708.1	86,874/AP019709.1	48,572/AP019710.1	[41]
95-3192	Clinical (USA, NA)	O145:H28	E/ST32	*stx* _2a_	5,385,516/CP027362.1	N/A	N/A	[42]
2015C-3125	Clinical (USA, 2014)	O145:H28	E/ST32	*stx*_1a_ + *stx*_2a_	5,471,132/CP027763.1	66,944/CP027764.1	66,388/CP027765.1	[42]
RM8843-C1	Cattle (USA, 2009)	O145:H28	E/ST32	*stx* _1a_	5,458,415/CP035772.1	88,752/CP035773.1	N/A	[30]
RM8988-C1	Cattle (USA, 2009)	O145:H28	E/ST32	*stx* _1a_	5,458,186/CP035770.1	88,752/CP035771.1	N/A	[30]
RM8995-C1	Sediment (USA, 2009)	O145:H28	E/ST32	*stx* _1a_	5,457,980/CP031355.1	88,747/CP031354.1	N/A	[30]
RM9154-C1	Cattle (USA, 2009)	O145:H28	E/ST32	*stx* _2a_	5,205,721/CP031353.1	86,711/CP031352.1	187,274/CP031350.1; 96,355/CP031351.1	[30]
RM9467-C1	Cattle (USA, 2009)	O145:H28	E/ST32	*stx* _2a_	5,385,895/CP031349.1	89,518/CP031348.1	N/A	[30]
RM9872-C1	Cattle (USA, 2009)	O145:H28	E/ST32	*stx* _2a_	5,385,904/CP024659.1	89,518/CP024660.1	N/A	[30]
RM9873-C1	Cattle (USA, 2009)	O145:H28	E/ST32	*stx* _2a_	5,385,819/CP031347.1	89,515/CP031346.1	N/A	[30]
RM10425-C1	Cattle (USA, 2009)	O145:H28	E/ST32	*stx* _2a_	5,343,037/CP031343.1	89,519/CP031342.1	N/A	[30]
RM11626-C1	Cattle (USA, 2010)	O145:H28	E/ST32	*stx* _1a_	5,419,044/CP035768.1	86,875/CP035769.1	N/A	[30]
RM12275-C1	Cattle (USA, 2010)	O145:H28	E/ST32	*stx* _1a_	5,408,598/CP031341.1	88,744/CP031340.1	N/A	[30]
RM12367-C1	Water (USA, 2010)	O145:H28	E/ST32	*stx*_2a_ + *stx*_2a_	5,472,396/CP031345.1	90,831/CP031344.1	N/A	[30]
RM12522-C8	Cattle (USA, 2010)	O145:H28	E/ST32	*stx* _1a_	5,418,923/CP035767.1	86,874/CP035766.1	N/A	[30]

^a^ The source and isolation year for strain MG1655 is based on the information available for its parental strain K-12 as described previously [43].^b^ The serotype of strain MG1655 is determined by BLAST searches of *E. coli* O-antigen and H-antigen databases described previously [44,45].^c^ pEHEC refers to the plasmid carrying genes encoding enterohemolysin.

**Table 2 microorganisms-10-00866-t002:** Characteristics of Pathogenicity Islands (PAIs) and Genomic Islands (GIs) examined in this study.

PAIs/GIs	Length (bp) /%GC	Sources of Query Sequences	GenBank Accession #/Positions	References
Locus of Enterocyte Effacement (LEE)	43,418/40.9	STEC O157:H7 str. EDL933	AE005174.2/4,649,862–4,693,279	[17]
Pathogenicity Island OI-122	23,455/46.3	STEC O157:H7 str. EDL933	AE005174.2/3,919,348–3,942,802	[17]
Pathogenicity Island OI-57	80,502/51.4	STEC O157:H7 str. EDL933	AE005174.2/1,849,324–1,929,825	[17]
Tellurite Resistance Island (TRI)	87,548/48.0	STEC O157:H7 str. EDL933	AE005174.2/1,454,242–1,541,789	[17]
Locus of Adhesion and Autoaggregation (LAA)	86,353/48.6	STEC O91:H21 str. B2F1	AFDQ01000026.1/385,984–472,336	[20]
Locus of Proteolysis Activity (LPA)	37,710/47.4	STEC O91:H^-^ str. 4797/97	AJ278144.1/1–37,710	[47]
High-Pathogenicity Island (HPI)	36,448/56.4	*Yersinia pestis*	AL031866.1/78,113–114,560	[48]
Subtilase Encoding Pathogenicity Island (SE-PAI)	8058/46.2	*E. coli* str. ED32	JQ994271.1/1–8058	[49]
Acid Fitness Island (AFI)	13,620/46.0	*E. coli* str. MG1655	U00096.3/3,653,961–3,667,580	[19]
Locus of Heat Resistance (LHR)	14,981/62.2	*E. coli* str. P12b	CP002291.1/319,821–304,841	[21]

**Table 3 microorganisms-10-00866-t003:** Genetic features of Pathogenicity Islands LEE, OI-122, and OI-57.

Strains	^a^ LEE			^b^ OI-122			^c^ OI-57		
	Integration Site	Chromosomal Location	Size (bp)/CDS	Integration Site	Chromosomal Location	Size (bp)/CDS	Integration Site	Chromosomal Location	Size (bp)/CDS
EDL933	*selC*	4,649,862–4,693,279	43,418/52	*pheV/pheU*	3,919,348–3,942,802	23,455/27	*yciD* (*ompW*)	1,849,324–1,929,825	80,502/107
CFSAN004176	*pheV*	2,363,880–2,425,764	61,885/60	*pheV/pheU*	3,818,649–3,780,408	38,242/40	NA	NA	NA
CFSAN004177	*pheV*	2,825,461–2,763,585	61,877/61	*pheV/pheU*	1,370,795–1,409,031	38,237/40	NA	NA	NA
RM13514	*selC*	4,558,936–4,605,536	46,601/55	*pheV/pheU*	5,269,766–5,222,118	47,649/47	*ompW*	1,591,159–1,633,674	42,516/51
RM13516	*selC*	4,410,578–4,458,362	47,785/57	*pheV/pheU*	5,188,896–5,242,157	53,262/53	*ompW*	1,569,959–1,615,548	45,590/55
10942	*selC*	4,428,998–4,475,832	46,835/56	*pheV/pheU*	5,060,248–5,047,349	12,900/10	*ompW*	1,553,610–1,597,236	43,627/50
112648	*selC*	4,538,738–4,585,714	46,977/56	*pheV/pheU*	5,203,887–5,157,474	46,414/37	*ompW*	1,631,815–1,715,209	83,395/102
122715	*selC*	4,436,958–4,483,934	46,977/56	*pheV/pheU*	5,103,184–5,055,469	47,716/39	*ompW*	1,575,808–1,619,718	43,911/50
95-3192	*selC*	2,903,392–2,950,270	46,879/55	*pheV/pheU*	3,567,963–3,521,794	46,170/33	*ompW*	506,141–491,757	14,385/24
2015C-3125	*selC*	4,369,007–4,416,023	47,017/55	*pheV/pheU*	5,029,590–4,987,764	41,827/34	*ompW*	1,444,864–1,459,247	14,384/24
RM8843-C1	*selC*	4,473,390–4,520,271	46,882/56	*pheV/pheU*	5,142,570–5,091,970	50,601/55	*ompW*	2,204,394–2,248,382	43,989/59
RM8988-C1	*selC*	4,473,157–4,520,039	46,883/56	*pheV/pheU*	5,141,855–5,091,255	50,601/55	*ompW*	1,628,125–1,613,741	14,385/26
RM8995-C1	*selC*	4,472,986–4,519,862	46,877/56	*pheV/pheU*	5,142,141–5,091,544	50,598/55	*ompW*	3,201,502–3,157,514	43,989/59
RM9154-C1	*selC*	4,237,174–4,284,056	46,883/56	*pheV/pheU*	4,891,189–4,855,660	35,530/36	*ompW*	159,456–186,929	27,474/44
RM9467-C1	*selC*	4,411,115–4,458,137	47,023/56	*pheV/pheU*	5,071,448–5,029,449	42,000/42	*ompW*	1,574,704–1,618,656	43,953/58
RM9872-C1	*selC*	4,411,123–4,458,145	47,023/58	*pheV/pheU*	5,071,457–5,029,079	42,379/44	*ompW*	2,099,371–2,084,987	14,385/24
RM9873-C1	*selC*	4,411,053–4,458,073	47,021/57	*pheV/pheU*	5,071,377–5,029,379	41,999/42	*ompW*	1,574,682–1,618,634	43,953/58
RM10425-C1	*selC*	4,368,259–4,415,280	47,022/56	*pheV/pheU*	5,028,591–4,986,592	42,000/42	*ompW*	2,099,365–2,084,981	14,385/26
RM11626-C1	*selC*	4,439,648–4,486,530	46,883/56	*pheV/pheU*	5,104,582–5,058,414	46,169/41	*ompW*	3,172,643–3,158,259	14,385/26
RM12275-C1	*selC*	4,472,938–4,519,814	46,877/56	NA	NA	NA	*ompW*	1,622,111–1,666,070	43,960/59
RM12367-C1	*selC*	4,497,622–4,544,644	47,023/56	*pheV/pheU*	5,157,953–5,115,955	41,999/42	*ompW*	1,575,949–1,713,896	137,948/191
RM12522-C8	*selC*	4,439,541–4,486,422	46,882/56	*pheV/pheU*	5,104,468–5,058,301	46,168/41	*ompW*	1,625,877–1,652,995	27,119/40

^a^ The LEE island in strain EDL933 corresponds to O-island #148 as reported previously [17]. The LEEs in strains RM13514 and RM13516 were reported previously [6]. The LEEs in strains 10942, 112648, and 122715 were based on the genome annotations reported previously [41]. The LEE regions in O145:H25 strains were defined by IslandViewer4 in this study. The LEE regions in environmental STEC O145:H28 strains were defined by BLASTn search of a database containing all STEC genomes examined in this study using the RM13514 LEE as a query. ^b^ The OI-122 PAIs in strains RM13514 and RM13516 were reported previously [6]. The OI-122 PAIs in strains 10942, 112648, and 122715 were based on the genome annotations reported previously [41]. The OI-122 regions in environmental STEC O145:H28 strains and in O145:H25 strains were defined by BLASTn search of a database containing all STEC genomes examined in this study using the OI-122 of the strain EDL933 and of the RM13514 as queries. The tRNA genes *pheV* and *pheU* are identical. In some genomes, this tRNA gene was annotated as *pheV* whereas in others, it was annotated as *pheU*. In strain RM12275-C1, this integration site is unoccupied. ^c^ The OI-57 regions in O145 strains were defined by BLASTn search of a database containing all STEC genomes examined in this study using the EDL933 OI-57 as a query.

**Table 4 microorganisms-10-00866-t004:** Genetic features of the Tellurite Resistance Island (TRI), Locus of Adherence and Autoaggregation (LAA), and Acid Fitness Island (AFI).

Strains	^a^ Tellurite Resistance Island (TRI)			Locus of Adherence and Autoaggregation (LAA) Module Location/Size (bp)				Acid Fitness Island (AFI)	
	Related tRNA Genes	Chromosomal Locations	Size (bp)/ CDS	I	II	III	IV	Chromosomal Locations	Size(bp)/ CDS
EDL933	OI-43/*serW*; OI-48/*serX*	OI-43:1,058,620–1,146,182	87,563/106	OI-43/864	OI-43/ 5777	OI-43/ 11,256	OI-43/15,455	4,454,268–4,476,943	22,676/21
		OI-48:1,454,242–1,541,789	87,548/105	OI-48/864	OI-48/ 5777	OI-48/ 11,255	OI-48/15,455		
CFSAN004176	*ileX*	3,679,064–3,626,942	52,123/61	NA	^b^ TRI/ 4994	NA	TRI/2486	3,171,287–3,157,668	13,620/13
CFSAN004176	*ileX*	1,510,368–1,562,482	52,115/60	NA	^b^ TRI/ 4995	NA	TRI/2486	2,018,117–2,031,735	13,619/13
RM13514	TRI-1/*serX*; TRI-2/*ileX*	TRI-1:1,241,544–1,322,994	81,451/103	TRI-1/863	TRI-1/ 5777 ^b^ TRI-2/ 4950	TRI-1/ 9945	TRI-1/15,593; OI-122/13,972; TRI-2/2486	4,360,251–4,382,890	22,640/24
		TRI-2:3,926,463–3,861,137	65,327/67						
RM13516	*serX*	1,206,887–1,303,036	96,150/111	TRI/863	TRI/5776	TRI/7083	TRI/14,084; OI-122/19,082; ^c^ IE06/19,939	4,224,613–4,247,252	22,640/24
10942	*serX*	1,202,851–1,284,087	81,237/97	TRI/863	TRI/5777	TRI/9945	TRI/14,142	4,243,061–4,265,700	22,640/20
112648	*serX*	1,280,897–1,362,131	81,235/97	TRI/863	TRI/5777	TRI/9696	TRI/14,140; OI-122/13,972	4,352,804–4,375,443	22,640/20
122715	*serX*	1,223,578–1,303,501	79,924/94	TRI/863	TRI/ 5777	TRI/ 9945	TRI/14,142; OI-122/15,273	4,251,163–4,273,802	22,640/20
95-3192	*serX*	856,722–775,651	81,072/86	TRI/862	TRI/ 5766	TRI/ 9944	TRI/14,141; OI-122/13,960	2,701,442–2,724,081	22,640/21
2015C-3125	*serX*	1,095,410–1,175,328	79,919/86	TRI/862	TRI/ 5775	TRI/ 9945	TRI/14,142; OI-122/15,218	4,183,005–4,205,643	22,639/21
RM8843-C1	*serX*	1,852,148–1,933,384	81,237/99	TRI/863	TRI/ 5777	TRI/ 9696	TRI/14,142; OI-122/17,759	4,287,369–4,310,008	22,640/22
RM8988-C1	*serX*	1,980,371–1,899,135	81,237/99	TRI/863	TRI/ 5777	TRI/ 9696	TRI/14,142; OI-122/17,759	4,287,136–4,309,775	22,640/22
RM8995-C1	*serX*	2,961,815–2,880,586	81,230/99	TRI/863	TRI/ 5777	TRI/ 9695	TRI/14,140; OI-122/17,758	4,286,971–4,309,610	22,640/22
RM9154-C1	*serX*	1,222,582–1,291,597	69,016/90	TRI/863	TRI/ 5777	TRI/ 9945	TRI/14,142; OI-122/3,488	4,036,160–4,058,799	22,640/22
RM9467-C1	*serX*	1,223,947–1,305,183	81,237/99	TRI/863	TRI/ 5777	TRI/ 9946	TRI/14,141; OI-122/15,224	4,225,103–4,247,742	22,640/22
RM9872-C1	*serX*	2,450,129–2,368,893	81,237/103	TRI/863	TRI/ 5777	TRI/ 9946	TRI/14,142; OI-122/15,224	4,225,111–4,247,750	22,640/24
RM9873-C1	*serX*	1,223,939–1,305,171	81,233/99	TRI/863	TRI/ 5777	TRI/ 9946	TRI/14,140; OI-122/15,223	4,225,047–4,247,685	22,639/22
RM10425-C1	*serX*	2,407,270–2,326,034	81,237/99	TRI/863	TRI/ 5777	TRI/ 9946	TRI/14,142; OI-122/15,224	4,182,248–4,204,887	22,640/22
RM11626-C1	*serX*	2,879,078–2,796,529	82,550/101	TRI/863	TRI/ 5777	TRI/ 9696	TRI/14,142; OI-122/13,972	4,253,636–4,276,275	22,640/22
RM12275-C1	*serX*	1,269,895–1,351,122	81,228/99	TRI/863	TRI/ 5777	TRI/ 9695	TRI/14,141	4,286,927–4,309,564	22,638/22
RM12367-C1	*serX*	1,223,885–1,305,121	81,237/99	TRI/863	TRI/ 5777	TRI/ 9946	TRI/14,142; OI-122/15,224	4,311,611–4,334,250	22,640/22
RM12522-C8	*serX*	1,274,000–1,356,546	82,547/101	TRI/863	TRI/ 5777	TRI/ 9696	TRI/14,142; OI-122/13,971	4,253,530–4,276,169	22,640/22

^a^ The TRI islands in strain EDL933 are OI-43 and OI-48 as reported previously [17]. The TRI regions in all O145:H28 strains were defined by BLASTn search of a database containing all STEC genomes examined in this study using the EDL933 OI-48 as a query. The TRI regions in O145:H25 strains were defined by BLAST search genome each using the TRI-2 (inserted downstream of the *ileX* gene in strain RM13514) as a query as no homologs were detected when OI-48 was used as a query. ^b^ This 5 kb LAA Module II segment in both O145:H25 strains mainly carries the virulence gene *lesP* (*espC*). A homolog of this segment is present on the virulence plasmid pEHEC. ^c^ IE06 is based on the annotation reported previously [6].

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
