# Peer review of "Comparative Genomics Applied to Systematically Assess Pathogenicity Potential in Shiga Toxin-Producing Escherichia coli O145:H28"

_microorganisms, 2022, doi:10.3390/microorganisms10050866_

Round 1
Reviewer 1 Report
A very interesting paper comparing genomic data from clinical and environmental strains of STEC O145:H28, which belong to one of the severe illness-related serogroups. The comparative analysis established a clear separation of strains of this serogroup and strains of serotype O157:H7 in terms of core genome, what is more they are able to differentiate among clinical and environmental strains when the accessory genome is analysed. My suggestion, aimed to improve the quality of the research, is to carry out expression studies. Currently there are a lot of papers on comparative genomics, which are very interesting and have the potential to identify virulence genes more often associated with clinical strains, but the prediction of clinical outcome will benefit from studies focused on expression of genes
Author Response
Please see the attachment.
Response 1: We agree with the reviewer that examination of gene expression would provide additional information in prediction of virulence/pathogenicity of STEC strains. In fact, we have performed the RNA-seq based transcriptomic study to identify the core set of virulence genes that are upregulated under conditions relevant to survival of STEC in human hosts. This is particularly interesting and would provide insightful information considering the widespread loss-of-function mutations in various virulence genes and in genes encoding transcriptional regulators. However, this research is beyond the scope of our current investigation. We will plan on submitting a future publication focusing on the examination of gene expression as a build-on study of the current research.
Reviewer 2 Report
This is a very detailed comparative genomic analysis of 19 STEC O145:H28 strains (7 from clinical isolates and 12 from environmental) and four reference strains based on 333 virulence genes belonging to 10 major functional categories which contribute to E. coli pathogenicity. The study provides extensive information about how the genes involved in pathogenesis vary among the strains and which are the main sources of this genetic variation. This information is very important in order to characterize as detailed as possible the STEC strains and may be develop tools to monitor their expansion, since they represent a major threat to Public Health.
A point I would like (as a reader) to be clarified is whether the genes analyzed are all part of the accessory genome or some are of the core genome as well.
Also, some corrections should be made to the following lines:
Line 94: “This divergencee in…”
Line 110: “potenetial in STEC environmental isolates”
Line 189: “reference geneoms were included in the analyses, as..”
Line 546: “outer memberane protein intimin, medi- ..”
Author Response
Please see the attachment.
Point 1: This is a very detailed comparative genomic analysis of 19 STEC O145:H28 strains (7 from clinical isolates and 12 from environmental) and four reference strains based on 333 virulence genes belonging to 10 major functional categories which contribute to E. coli pathogenicity. The study provides extensive information about how the genes involved in pathogenesis vary among the strains and which are the main sources of this genetic variation. This information is very important in order to characterize as detailed as possible the STEC strains and may be develop tools to monitor their expansion, since they represent a major threat to Public Health.
Response 1: We thank the reviewer for the positive comments regarding the significance of our study.
Point 2: A point I would like (as a reader) to be clarified is whether the genes analyzed are all part of the accessory genome or some are of the core genome as well.
Response 2: We included this information in the Supplemental Table S1 in the revised manuscript.
Point 3: Also, some corrections should be made to the following lines:
Line 94: “This divergencee in…”
Line 110: “potenetial in STEC environmental isolates”
Line 189: “reference geneoms were included in the analyses, as..”
Line 546: “outer memberane protein intimin, medi- ..”
Response 3: We thank the reviewer for identifying the above typos. We corrected all in the revised manuscript ( Lines 94 (divergence), 110 (potential), 190 (genomes), and 546 (membrane)). We also corrected additonal typos and grammar errors in the revised manuscirpt (highlighted in yellow).